# Colombia's bioregions as a source of useful plants

**Nadia Bystriakova**[1⊙]*, **Carolina Tovar**[2⊙], **Alexandre Monro**[2], **Justin Moat**[2], **Pablo Hendrigo**[3], **Julia Carretero**[2], **Germán Torres-Morales**[4], **Mauricio Diazgranados**[2]

**1** Core Research Laboratories, The Natural History Museum, London, United Kingdom, **2** Royal Botanic Gardens, Kew, London, United Kingdom, **3** Centro Nacional de Conservação da Flora, Instituto de Pesquisas Jardim Botânico do Rio de Janeiro, Rio de Janeiro, RJ, Brazil, **4** Instituto de Investigación de Recursos Biológicos Alexander von Humboldt, Bogotá, Colombia

⊙ These authors contributed equally to this work.
* n_bystriakova@yahoo.com

**Data Availability Statement:** All relevant data are available on figshare: https://doi.org/10.6084/m9.figshare.15147402.

## Abstract

The aim of our study was to assess the importance of different Colombian bioregions in terms of the supply of useful plant species and the quality of the available distribution data. We assembled a dataset of georeferenced collection localities of all vascular plants of Colombia available from global and local online databases. We then assembled a list of species, subspecies and varieties of Colombia's useful plants and retrieved all point locality information associated with these taxa. We overlaid both datasets with a map of Colombia's bioregions to retrieve all species and useful species distribution records in each bioregion. To assess the reliability of our estimates of species numbers, we identified information gaps, in geographic and environmental space, by estimating their completeness and coverage. Our results confirmed that Colombia's third largest bioregion, the Andean moist forest followed by the Amazon, Pacific, Llanos and Caribbean moist forests contained the largest numbers of useful plant species. Medicinal use was the most common useful attribute across all bioregions, followed by Materials, Environmental uses, and Human Food. In all bioregions, except for the Andean páramo, the proportion of well-surveyed 10×10 km grid cells (with ≥ 25 observation records of useful plants) was below 50% of the total number of surveyed cells. Poor survey coverage was observed in the three dry bioregions: Caribbean deserts and xeric shrublands, and Llanos and Caribbean dry forests. This suggests that additional primary data is needed. We document knowledge gaps that will hinder the incorporation of useful plants into Colombia's stated plans for a bioeconomy and their sustainable management. In particular, future research should focus on the generation of additional primary data on the distribution of useful plants in the Amazon and Llanos (Orinoquia) regions where both survey completeness and coverage appeared to be less adequate compared with other regions.

**Funding:** This work is supported by a Professional Development & Engagement grant under the Newton-Caldas Fund partnership. The grant is funded by the UK Department for Business, Energy and Industrial Strategy (BEIS) and the Colombian Ministry of Science, technology and Innovation (MinCiencias), and delivered by the British Council. For further information, please visit www. newtonfund.ac.uk - Initials of the authors who received each award MD - Grant numbers awarded to each author N/A - The full name of each funder Newton-Caldas Fund Colombian Ministry of Science, Technology and Innovation (MinCiencias) British Council - URL of each funder website www. newtonfund.ac.uk https://www.britishcouncil.org/ https://minciencias.gov.co/ The funders had no role in study design, data collection and analysis, decision to publish, or preparation of the manuscript.

**Competing interests:** The authors have declared that no competing interests exist.

## Introduction

With 26,134 plant species documented [1], the flora of Colombia is the second most diverse within the Americas [2, 3]. Plants play a central role in the supply of ecosystem services of water, provisioning, and nutrient cycling [4]. For local communities, plants are important sources of human and animal food, medicines, building materials, fuel and culture [5, 6]. However, as in the vast majority of tropical countries, traditional knowledge about plants in Colombia is under-documented [7]. This is despite Colombia being considered the "cradle of modern ethnobotany", due to the research of several ethnobotanists across the country in the last decades [8]. In general, Colombia has a long history of ethnobotanical studies that have covered most of its territory, except for the Orinoquia [8]. These studies provide information not only on plant taxonomy and distribution, but also on traditional uses of plants by local communities. The high diversity of Colombian plants is matched by the diversity of its ecosystems [9], some of which are of global conservation importance, notably high elevation tropical alpine ecosystems (páramos), the Chocó and Andean forests [2, 10, 11]. The importance of different ecosystems as sources of useful plants for Colombia has not been evaluated in a comparable way, the knowledge base being the result of independent and unconnected studies limited to single regions or subregions [12–14].

Whilst Colombia's ecosystems deliver numerous services that underpin the wellbeing of Colombians, they are also sensitive to environmental change and difficult to restore, because of their high diversity and complexity [15–17]. Given a very high risk of plant extinction in the Anthropocene [18], potentially useful plant species might be lost to climate and land use change even before their uses have been recognised. Also, they are under increased pressure from unsustainable land-use practices following the end of a decades-long civil war [19, 20]. Within the current post-war era, Colombia has stated its intention to develop as a bio-economy, that is "[an] economy that efficiently and sustainably manages biodiversity and biomass to generate new value-added products, processes and services based on knowledge and innovation", through its "Política de Crecimiento Verde–Green Growth Policy" [21]. Colombia expects its bioeconomy to, in a large part, rely on the sustainable management of its vascular flora in general and on plants with known uses, referred to hereafter as 'useful plants', in particular. It is fundamental to this aim that a complete evaluation of useful plant information and of gaps in this knowledge be identified with respect to taxa, ecosystems, geography and people.

Inadequate sampling effort is a common problem and a major obstacle to a better understanding of ecosystem function globally [22]. Despite recent advances in remotely sensed estimates of species richness [23], biological collections metadata and plot-data combined remain the major source of information on diversity and distribution of terrestrial vegetation. Undertaking comprehensive plant surveys are particularly challenging in mega-diverse countries, such as Colombia, where funding and access to the field is limited, creating barriers for better sampling. Identifying sampling gaps is important, as it supports the interpretation of biodiversity analyses [24], but also can provide a framework for prioritising future investment in biodiversity exploration.

The aim of the present study was to 1) assess the importance of different ecosystems for the supply of useful plant species and 2) evaluate the quality of the primary data on which this is based (i.e. distribution data from georeferenced records). In order to provide a broader context to our results, we focused our study on useful plants as a subset of all Colombian vascular plants. Because the Colombian Ecosystem map [9] with its 93 ecosystem types was too detailed for the purposes of our study, we produced our own map with 14 generalized ecosystem types (bioregions) that includes Colombia main geographic regions and biomes. We ask the

following specific research questions: 1. Which Colombian bioregions have the greatest diversity of useful plant species? 2. What are the most observed plant use categories across Colombia's bioregions? 3. How well have Colombian bioregions been sampled?

## Materials and methods

To answer the above questions, we assembled a dataset of georeferenced collection localities (hereafter distribution records) of all vascular plants of Colombia available from online databases. We also built a useful plants list for Colombia and retrieved all available information on the distribution of useful plants from the dataset of all vascular plants. Reviewing the Colombian Ecosystem map (IDEAM et al., 2017) with its 93 ecosystem types, we found it too detailed for the purposes of our study. We produced our own bioregion map with 14 units that includes Colombia main geographic regions and biomes. We overlaid both datasets with the bioregion map to assess species richness and number of distribution records in each bioregion. Lastly, to assess the reliability of our estimates of species richness and inform future sample efforts, we identified information gaps in the available plant distribution records, in geographic and environmental space, by estimating their completeness and coverage.

### Datasets

**Distribution records.** The distribution records for all vascular plants of Colombia were downloaded from the SiB Colombia open national biodiversity data network and GBIF in February 2020 [25]. In order to mitigate some of the known limitations of species occurrence data [22, 26], names were checked against the Plants of the World online (POWO) using the Python interface to Kew data (PyKew: https://pypi.org/project/pykew/), and geographic coordinates were cleaned using CoordinateCleaner [27] in R (S1 File). The resulting dataset comprised 522,257 unique observations of 23,961 species from 281 families (S2.1 Table and S2.1 Fig in S2 File).

The most frequently recorded species was *Gaultheria myrsinoides* (1015 unique records), and there were 4,721 "point endemics", i.e. species recorded at a single location (19.7% of all species). On average, there were 21.79 records per species (S2.1 Table in S2 File).

**Colombian useful plants list.** The dataset of all useful vascular plants of Colombia was obtained from the Useful Plants and Fungi (UPFC) project (https://www.kew.org/upfc). The species list was obtained by compiling reports of plant use from more than ten online datasets (e.g., http://i2d.humboldt.org.co/ceiba/) and peer-reviewed publications [28, 29]. The dataset used for this study comprised 4,230 vascular plant species (UPFC project up to July 2020), including 3,589 (85%) species with confirmed uses in Colombia and 644 (15%) species reported by both the World Checklist of Useful Plants 2020 [28] and the Colombian Catalogue of Plants and Lichens [2], but without confirmed use in Colombia yet. The nomenclatural reconciliation of the names was carried out using the taxonomic backbone from Plants of the World Online (POWO; http://www.plantsoftheworldonline.org/) and ColPlantA (https://colplanta.org/). A few unresolved names were checked against Tropicos 3.0.2 (23 names; https://www.tropicos.org/) and the Global Biodiversity Information Facility (GBIF) using species matching tool (30 names; https://www.gbif.org/tools/species-lookup). R packages plyr version 1.8.5 [30], rgdal version 1.4–8 [31] and doBy version 4.6–3 [32] were used for the reconciliation of scientific names. Misspelt species names were corrected as much as possible, and mismatched names were discarded. Synonymy and homonymy were resolved, and illegitimate and invalid names were excluded. IPNI Life Sciences Identifier (LSID) were assigned to most names (99%).

**Table 1. Categories of plant uses following Diazgranados et al. (2020).**

| Category of use | Description |
|---|---|
| ANIMAL FOOD | Forage and fodder for vertebrate animals only. |
| ENVIRONMENTAL USES | Examples include intercrops and nurse crops, ornamentals, barrier hedges, shade plants, windbreaks, soil improvers, plants for revegetation and erosion control, wastewater purifiers, indicators of the presence of metals, pollution, or underground water. |
| FUELS | Wood, charcoal, petroleum substitutes, fuel alcohols, etc.—have been separated from MATERIALS because of their importance. |
| GENE SOURCES | Wild relatives of major crops which may possess traits associated to biotic or abiotic resistance and may be valuable for breeding programs. |
| HUMAN FOOD | Food, including beverages, for humans only. |
| INVERTEBRATE FOOD | Only plants eaten by invertebrates useful to humans, such as silkworms, lac insects and edible grubs, are covered here. |
| MATERIALS | Woods, fibres, cork, cane, tannins, latex, resins, gums, waxes, oils, lipids, etc. and their derived products. |
| MEDICINES | Both human and veterinary. |
| POISONS | Plants which are poisonous to vertebrates and invertebrates, both accidentally and usefully, e.g. for hunting and fishing. |
| SOCIAL USES | Plants used for social purposes, which are not definable as food or medicines, for instance, masticatories, smoking materials, narcotics, hallucinogens and psychoactive drugs, contraceptives and abortifacients, and plants with ritual or religious significance. |

The dataset of all vascular plants of Colombia (see the above section) contained 197,166 unique distribution records of 3,870 species of useful plants (S2.1 Table and S2.2 Fig in S2 File). Plant uses were defined following Diazgranados et al. [28], which is a simplified version of that proposed by Cook [6], see Table 1 for definitions and Table 2 for the distribution of records and species across use categories.

With 877 unique records, *Quercus humboldtii* was identified as the most frequently recorded useful species. Only 221 (5.71%) of useful species were known from a single record/locality (S2.1 Table in S2 File). On average, there were 50.95 records per species. In our list of useful plants, 330 species (7.9%) did not have any geographic coordinates associated with them.

**Bioregions and environmental data.** In order to build a biome-region map (Fig 1A) we first used the terrestrial ecoregions map of the world [33] and identified the main biomes of

**Table 2. Useful plants and their uses.**

| Uses | Number of species | % of the total number of useful species | Number of records | % of the total number of records | Records:Species ratio |
|---|---|---|---|---|---|
| Medicines | 2954 | 76.33 | 159401 | 80.85 | 53.96 |
| Materials | 1475 | 38.11 | 86161 | 43.70 | 58.41 |
| Environmental Uses | 1233 | 31.86 | 56441 | 28.63 | 45.78 |
| Human Food | 1004 | 25.94 | 54365 | 27.57 | 54.15 |
| Animal Food | 609 | 15.74 | 33268 | 16.87 | 54.63 |
| Gene Sources | 541 | 13.98 | 25424 | 12.89 | 46.99 |
| Poisons | 469 | 12.12 | 24647 | 12.50 | 52.55 |
| Social Uses | 339 | 8.76 | 21969 | 11.14 | 64.81 |
| Fuels | 235 | 6.07 | 16924 | 8.58 | 72.02 |
| Invertebrate Food | 144 | 3.72 | 10807 | 5.48 | 75.05 |

Species and record numbers do not add up to the totals in S2.1 Table in S2 File, because species can have multiple uses; e.g. the same plant can be used both as Medicine and Human Food.

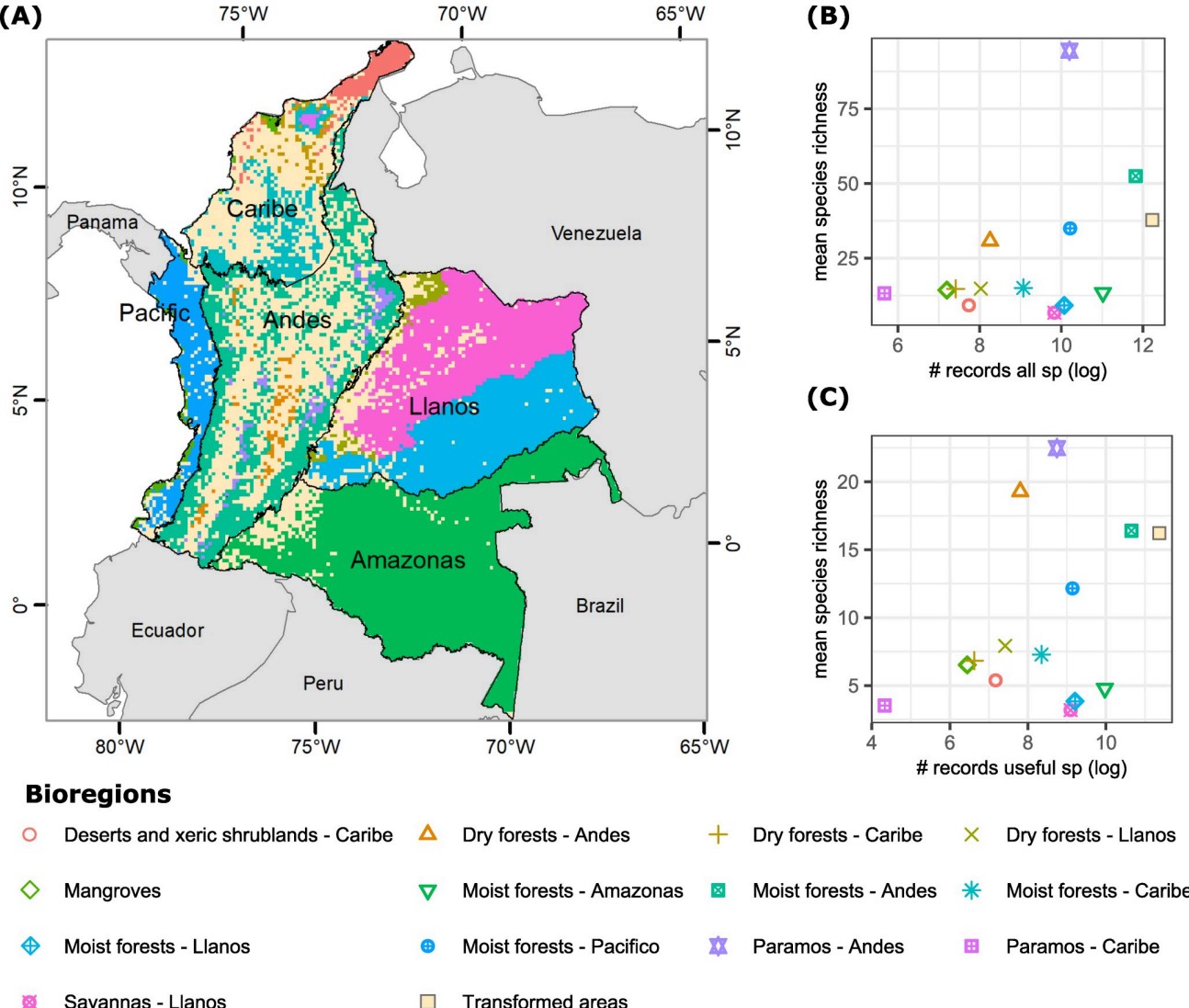

**Fig 1. Number of records of all vascular plants and useful species in bioregions.** (A) Bioregion map of Colombia. (B) Number of records of all vascular plants (log-transformed) vs the mean vascular species richness (average of the number of all vascular plant species found in each grid cell of 10 x 10 km) for each bioregion. (C) Number of records of useful plants (log-transformed) vs mean useful species richness (average number of useful species found in each grid cell of 10 x 10km. In (B) and (C), points at the top right of the plot represent bioregions with greater numbers of records, but also with higher numbers of species after accounting for area, because mean species richness (species/cells ratio) represents the average number of species per unit area (in this case, 10×10 km grid cell).

Colombia: Desert and xeric shrublands, Savannas, Páramos, Dry forests, Moist forests and Mangroves (S2.2 Table in S2 File). We checked updates of this map with its latest version [34] and verified no changes were needed for Colombia. Secondly, we used Colombia's regional map from the Instituto Geográfico Agustín Codazzi (IGAC) that identifies five main geographic regions: Caribbean, Pacific, Andes, Llanos and Amazon. Lastly, we overlapped those two maps to obtain our final 13 analysis units (S2.4 Table in S2 File). The shapefile was reprojected from GCS_WGS_84 (Datum WGS84) to South_America_Lambert_Conformal_Conic (Datum D_South_American_1969) (WKID 102015) and then converted to raster (10×10 km pixel size) in QGIS.

Because a large part of these ecosystems has been transformed, we used the Colombian Ecosystem map [9] downloaded from http://www.siac.gov.co/catalogo-de-mapas to create a mask of these regions. We used the following categories of the field "synthetic ecosystem" (ecosistemas sintéticos in Spanish): Transformed transitional, Artificial areas, and Agroecosystems. The shapefile of those categories was also reprojected from GCS_MAGNA (Datum D_MAGNA) to South_America_Lambert_Conformal_Conic (Datum D_South_American_1969) and then converted to raster (10×10 km pixel size).

The dataset of spatially interpolated monthly climate data (version 2.1) for global land areas at 5 minutes (approximately 10 km) spatial resolution [35] was downloaded from https://worldclim.org. In the absence of an *a priori* knowledge of the relationship between climate and vegetation of Colombia, we selected two variables, annual mean temperature (ANMT) and total annual precipitation (ANP) as those approximating the basic requirements of plants for energy and water.

## Analyses

All analyses were performed in RStudio Version 1.3.1093 (2009–2020 RStudio, PBC).

**Ecosystem analyses.** To identify the best sampled and most species rich bioregions we estimated sampling effort (measured as the total number of unique collection localities per bioregion) and the numbers of all vascular and useful plant species in the Colombian bioregions using the simplified raster with 10×10 km grid cells. We also estimated the numbers of collection localities in each category of use, across bioregions.

Because transformed ecosystem types (i.e. Transformed transitional, Artificial areas, and Agroecosystems) were not evenly distributed across the study area (S2.3 Table in S2 File), we merged all transformed areas into a single category (ID 14) and treated it as a bioregion in its own right for the purposes of the ecosystem analyses.

To study the effect of sampling effort on estimates of all vascular and useful plant species richness, defined as species to cell ratio, we created two subsets: all surveyed 10×10 km grid cells (i.e. those with at least one observation record) and well surveyed grid cells (i.e. those with ≥ 25 observation records). For each bioregion, we calculated mean species richness in all surveyed and well surveyed grid cells (i.e. the average of number of species in all grid cells included in the analysis).

**Survey completeness.** Sampling effort was measured as the total count of all unique collection localities and taxa, respectively, within 10×10 km grid cells. This resolution has been commonly used in regional gap analysis studies [24].

As a measure of survey completeness, we used numbers of surveyed (i.e. those with occurrence records) 10×10 km grid cells. In their country-scale analysis, Troia and McManamay [24] defined well-surveyed cells as those with ≥10, 25 and 50 occurrence records for the low, moderate and high thresholds, respectively. For purpose of our study, we defined well-surveyed cells in accordance with the moderate threshold, i.e. as those with ≥25 occurrence records, but it should be realised that this does not approach a truly well-sampled survey. This might be a generous estimate of the moderate threshold, because Troia and McManamay [24] worked on a temperate set of ecosystems, while our study was focused on tropical ecosystems which in general have more species.

**Survey coverage.** Survey coverage is commonly estimated along spatial, environmental and temporal gradients [24]. To find out whether sampling effort for all groups of organisms studied was adequate in relation to environmental gradients represented by present-day climatic variables (ANMT and ANP), we used a type of probability density function, kernel. A kernel is a special type of probability density function (PDF) with the added property that it

must be even. Thus, a kernel is a function with the following properties: it is non-negative, real-valued, even, and its definite integral over its support set must equal to 1. Some common PDFs are kernels; they include the uniform (-1, 1) and standard normal distributions [36]. Kernel density estimation is a non-parametric method of estimating the PDF of a continuous random variable. It is non-parametric because it does not assume any underlying distribution for the variable. Essentially, at every datum, a kernel function is created with the datum at its centre—this ensures that the kernel is symmetric about the datum. The PDF is then estimated by adding all of these kernel functions and dividing by the number of data to ensure that it satisfies the two properties of a PDF: every possible value of the PDF is non-negative; and the definite integral of the PDF over its support set equals to 1.

We estimated kernel density of the two continuous random variables representing Colombian climate at 5 minutes spatial resolution: annual mean temperature (ANMT), and annual precipitation (ANP). We then visually compared those distributions with kernel density of sampling efforts for all vascular plants and useful plants. We repeated the exercise for all vascular plants and useful plants in the four top species-rich bioregions. We would expect the PDFs of sampling effort and climatic variables to be similar in shape, if the group of living organisms studied has been adequately sampled across the region.

## Results

The largest bioregions, Andean, Amazon, Pacific, Llanos and Caribbean moist forests contained the highest numbers of species, of all vascular plants and useful plants. Across all ecosystems, vascular plants used as medicine and materials were recorded most often. In all bioregions, except for the Andean páramos, the proportion of well surveyed cells in the pool of all cells with useful plant records was below 50%.

### Most species-rich bioregions

With a few exceptions, the highest numbers of all vascular plant species and useful species and records were observed in the bioregions with the greatest extent: Andean, Amazon, Pacific, Llanos and Caribbean moist forests (Fig 1 and S2.4 Table in S2 File). Although the Pacific moist forest was less than half the size of the moist forest of Llanos, it had a greater diversity of all vascular plants and useful plants than the latter. Caribbean moist forest and savanna (Llanos) had nearly the same numbers of all vascular plant species and useful species, although the size of the former was only 8.7% of the latter.

Mean species richness of all vascular plants across bioregions was higher in the well surveyed areas compared with surveyed grid cells and the study area as a whole (Fig 2 and S2.5 Table in S2 File). With 147.8 species per 10×10 km grid cell, the Andean páramo (ID 3) was the most species rich bioregion across all extents closely followed by the Andean and Pacific moist forests (ID 10 and ID 13) when only well surveyed cells were considered. The Andean páramo (ID 3) appeared to be the most species rich bioregion across all study extents, while the Caribbean páramo (ID4) was among the bioregions with the lowest numbers of species per 10×10 km grid cell.

When useful plants were considered (Fig 2 and S2.6 Table in S2 File), the highest mean richness in the well surveyed cells was observed in the Amazon moist forest (ID 9, 67.3) followed by the Pacific and Andean moist forests (ID 13 and 10; 65.4 and 61.5 respectively). Across all extents, the lowest mean richness of useful plants was observed in the Caribbean páramo (ID 4).

In Transformed areas (ID 14), the values of mean species richness of all vascular plants and useful plants in well surveyed grid cells were among the highest across the bioregions.

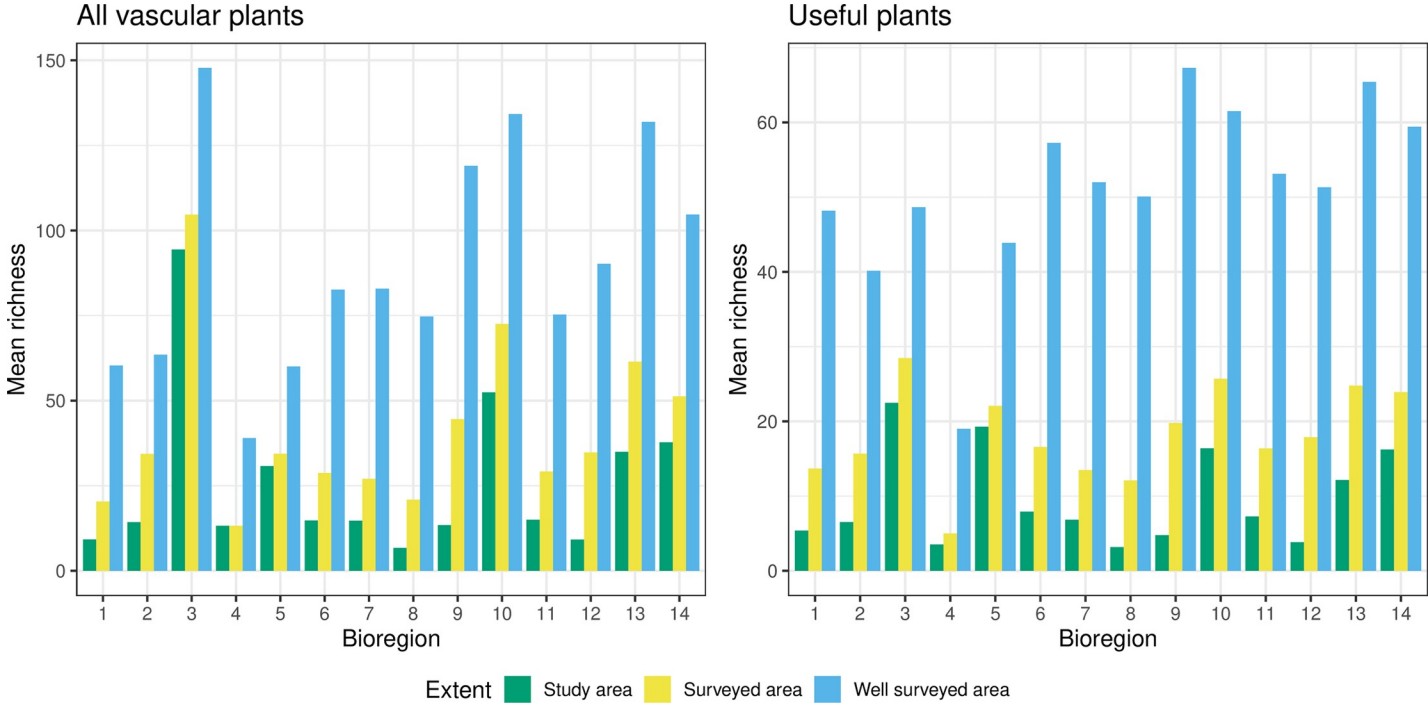

**Fig 2.** Mean species richness in the study area and in the surveyed areas: all vascular plants (A) and useful plants (B). Mean species richness in the study area was estimated as the number of species recorded in each 10 x 10 km grid cell within the geographic extent of Colombia; surveyed area richness was estimated within all 10 x 10 km grid cells with at least 1 record, and well surveyed area richness was estimated within all 10 x 10 km grid cells with ≥ 25 records. See Fig 1 for bioregion names and S2.5 and S2.6 Tables in S2 File, for richness values.

However, we could not estimate the degree of transformation within those areas; estimates of species richness could have been biased and are not comparable with other results.

### The best sampled categories of plant uses across bioregions

Across bioregions, plants used as medicines, materials, those with environmental uses, and animal food were recorded most often in all bioregions (Fig 3B and S2.7 Table in S2 File). We had no data on the number of individuals or biomass of useful species in the study area. More-over, we did not have any data on the actual use of plants across bioregions. Therefore, the results presented reflect potential rather than actual importance of different use categories in different bioregions.

The four top families with most useful species in Colombia across useful categories were first Fabaceae followed by Arecaceae, Euphorbiaceae and Malvacea (S2.8 Table in S2 File).

### Plant data quality

The quality of the plant data was assessed by survey completeness and coverage indices, including the number and percentage of all surveyed and well surveyed cells, and distribution of collections across selected environmental gradients.

**Survey completeness.** The largest proportions of well surveyed grid cells (i.e. with ≥ 25 observations) with records of all vascular plants were observed in the three Andean bioregions: páramo (69.2%), and moist and dry forests (51.8% in each, Fig 4A and S2.9 Table in S2 File). Moreover, in two of them (páramo and dry forest) ca 90% of the study area (i.e. all 10×10 km grid cells) have been surveyed. In the Llanos region, with 22.7% of well surveyed cells, savanna

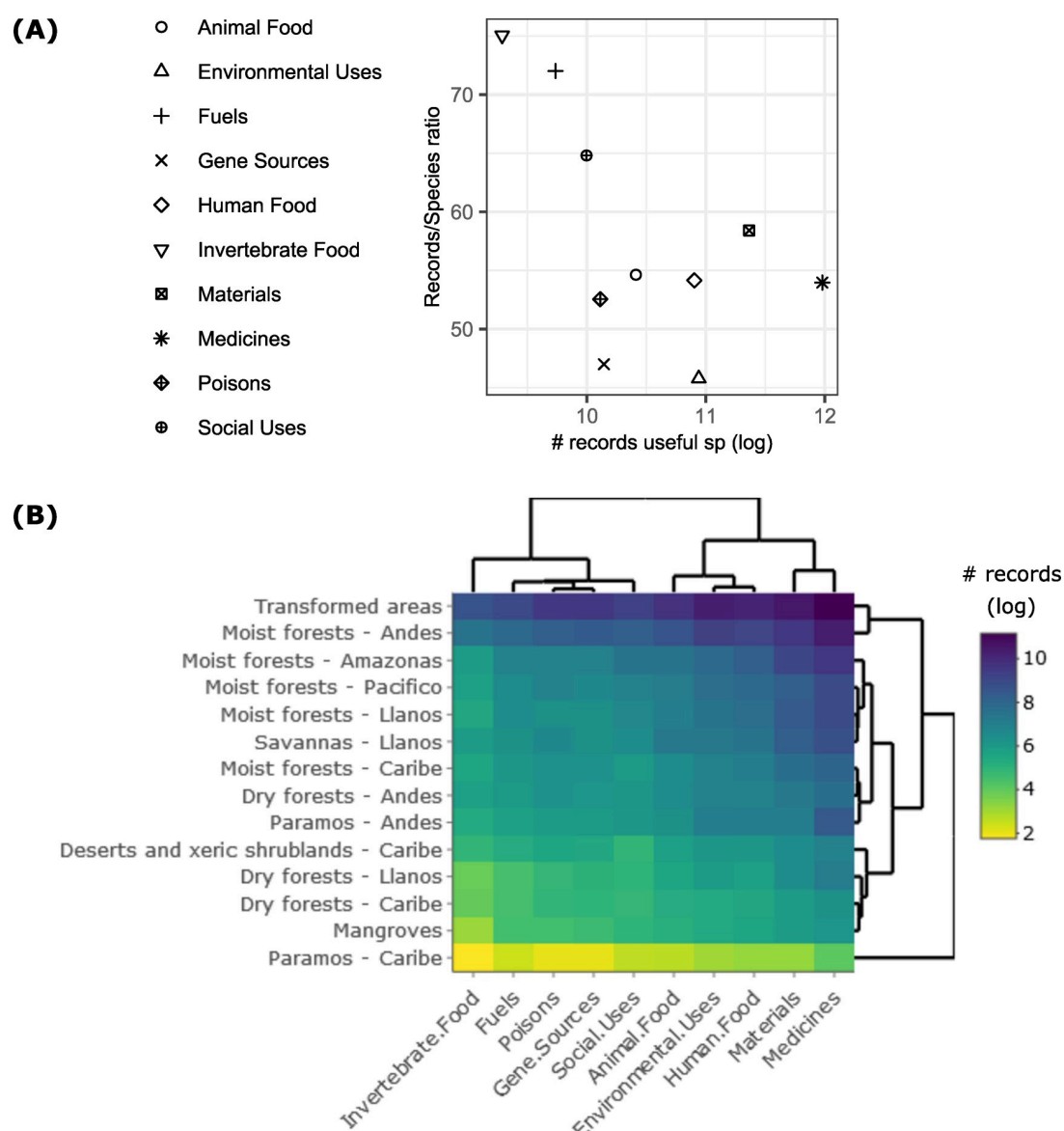

**Fig 3. Records of useful plants per category of use.** (A) Number of records of useful plants (log-transformed) vs the ratio of number of useful species per cell (species/cells ratio) for each category of use. (B) Number of records of useful plants (log-transformed) per category of use (axis x) and ecosystem (y axis). Dendrograms show clusters based on the number of records.

had the lowest survey completeness. While 100% of the Caribbean páramo grid cells have been surveyed, only 23.5% of those had 25 observations or more. The least surveyed bioregions were savanna (Llanos) with only 32% of all its area surveyed, and the Amazon and Llanos moist forests with 30.1% and 26.4%of their area surveyed, respectively.

When useful plants were considered, survey completeness was generally lower than in all plants (Fig 4B and S2.10 Table in S2 File). As it was the case with all plants, the Andean páramo was the best surveyed bioregion for useful plants; 78.9% of all cells were surveyed, and 52.4% of those had 25 observations or more. The lowest share of well surveyed cells was found in the Caribbean páramo, 8.3%.

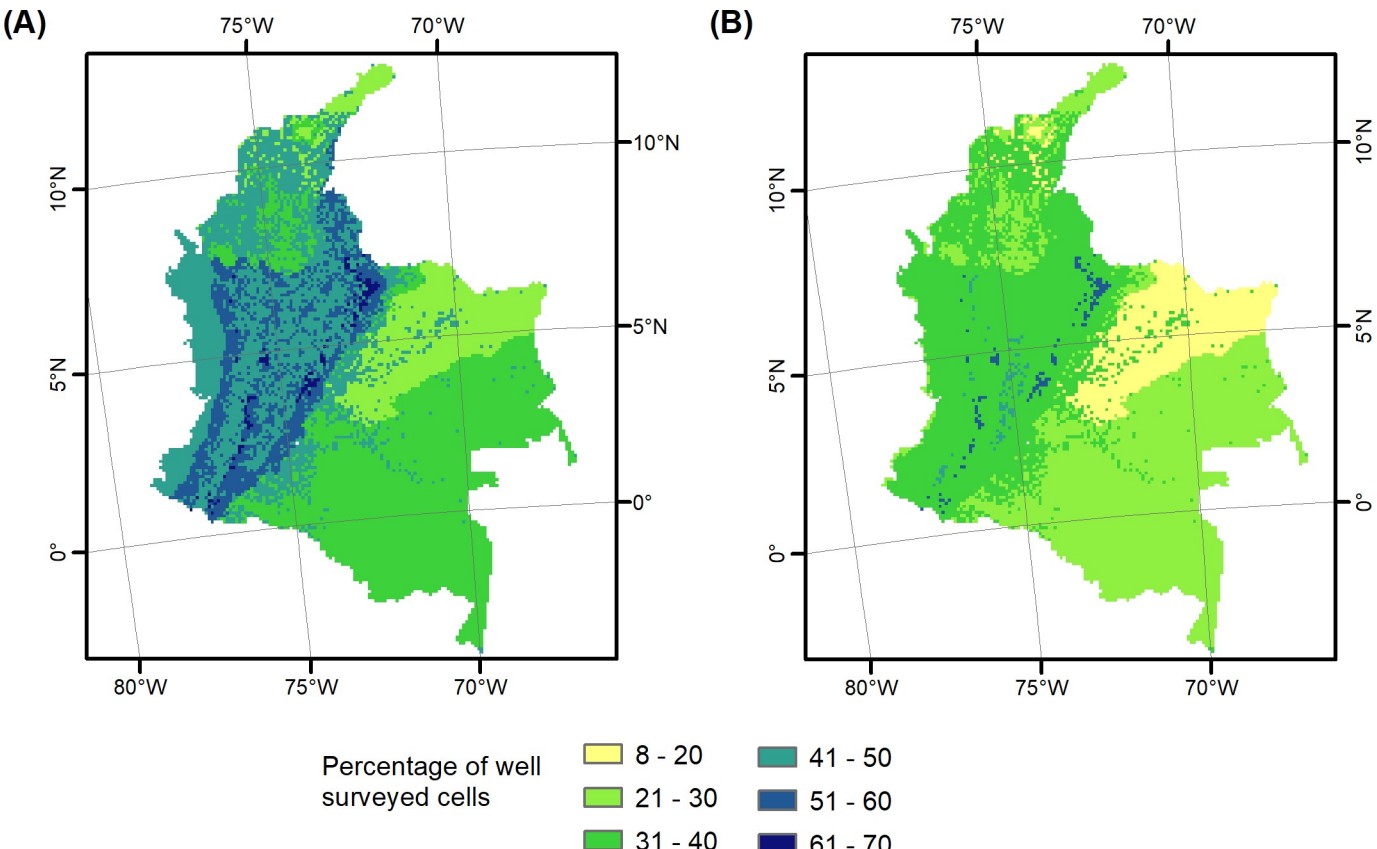

**Fig 4.** Survey completeness for all vascular (A) and useful (B) plants by bioregion. Percentage of 10×10 km well surveyed grid cells, defined as those with ≥ 25 observations, is calculated by bioregion.

**Survey coverage.**   We would expect the PDFs of sampling effort and climatic variables to be similar in shape, if the group of plants studied has been adequately sampled across the region. The results of visual examination of distribution of useful plants along temperature and precipitation gradients (S2.4 and S2.5 Figs in S2 File) suggest that survey coverage of useful plants compared to all plants is inadequate in some of the bioregions. In particular, poor survey coverage was observed in the three dry bioregions: Caribbean deserts and xeric shrublands (ID1), and Llanos and Caribbean dry forests (ID 6 and 7). In general, useful plants of Llanos and Amazonia appeared to be less adequately represented by collection localities compared with the Andean ecosystems. Of all ecosystem types, páramos were the most adequately sampled (ID 3 and 4).

## Discussion

Colombia is among the 41 nations that have adopted political strategies to promote bioeconomy [37] aimed to efficiently use and protect the countries' natural capital. Among the milestones to be achieved by 2030 are: increasing the number of bioeconomy start-ups by 180% and doubling the forest GDP to reach 2% of the total GDP [38]. In this context, increasing our knowledge about Colombia's useful plants distribution will contribute to the development of natural capital management strategies. According to the results of our study, nearly 4,000, or 16% of all Colombia's vascular plant taxa have at least one type of usage. It is important to note that this does not necessarily reflect numbers of species used in Colombia but the diversity of

species that are useful somewhere across their range. It is therefore more an estimate of potential, rather than, realised resource. The regions with the largest numbers of those taxa should be considered the priority resource for the new Colombia's economic growth model.

## Bioregions with the greatest numbers of useful species

In the following subsections we discuss each of the five Colombia's regions in terms of all vascular and useful plant species richness and information gaps identified by our study.

**The Andean region.** The Andean region is considered the most biodiversity rich in Colombia [15]. In our study it was subdivided into three bioregions: Andean moist and Andean dry forests, and Andean Páramos.

With 13,909 species of vascular plants and 2,781 species of useful plants, the Andean moist forest was the most species rich of all bioregions described in our study. It also had high numbers of vascular plants and useful plants per 10×10 km grid cell, 134.2 and 61.5 in the well surveyed area. Over 60% of the surveyed area and 35% of the well surveyed area had observations of useful plants. In our study, useful plants of the Andean moist forest were relatively well represented by collection localities in the climatic space.

With the area of ca 9,500 km$^2$, the Andean dry forest was among the smallest bioregions studied. Nevertheless, it had more species of vascular plants and useful plants than some of the bioregions with the larger area, including the dry forest of the Llanos. According to our estimates (S2.3 Table in S2 File), over 70% of the Andean dry forest area has been transformed by human activities.

The Andean Páramos are the high mountain ecosystems, located between the upper limit of the Andean forest, and the lower limit of the glaciers or perpetual snow [39]. Although Páramos provide numerous ecosystem services, including biodiversity, water regulation and soil stability, their sustainability is threatened by poor agricultural and livestock farming practices [40]. In our study the Andean Páramos were represented by 3,955 species of vascular plants and 816 useful species. The number of vascular plants was higher than reported by other authors [15, 39]. In our study, the Andean Páramos was the most species rich of all bioregions when the number of all vascular plants in 10×10 km grid cells was considered, with the mean of 148 in the well surveyed area. This region is also the most species rich of all tropical alpine regions, due to high net diversification rates likely associated with the Andean uplift [41, 42]. The records of vascular plants and useful plants of the Andean Páramos were found in 90% and 79% of all 10×10 km grid cells of the bioregion respectively. Over 69% and 52% of those cells had 25 or more observation records of vascular plants and useful plants, respectively. Thus the Andean Páramos had the highest survey completeness across all bioregions. Survey coverage for useful plants was adequate.

**Llanos (Orinoquia).** In our study, the Orinoquia region was represented by savanna, and the moist and dry forests of the Llanos. Despite being the second (after Amazon most forests) largest bioregion, with 2,938 species of vascular plants and 1,151 useful plants savanna ranked seventh in terms of species richness. This was consistent with the estimates of Rangel-Ch. [15] who described Orinoquia as the least species rich region of Colombia with ca 4,500 vascular plant species. Savanna also had the lowest proportion of well surveyed cells for vascular plants (22.7%) and the second lowest proportion for useful plants (16.3%), and rather poor survey coverage for useful plants. Inadequate sampling is likely the result of low human presence in savanna, where the relative size of transformed areas is ca 12%. It can also be due to a limited access to the region which includes both permanent and seasonally flooded savannas [43]. In the last 60 million years, hydrological changes have been a major driver of the region's transformation and likely shaped the present-day flora [44]. We suggest that species numbers in the

region are likely to have been underestimated by earlier studies, and would possibly exceed those reported by our project if better survey completeness and coverage could be achieved.

**Amazon moist forest.** In the Amazon moist forest, our study identified 6,650 species of vascular plants including 1,612 useful plants making this bioregion the second most species rich after the Andean moist forest. However, when well surveyed areas are considered the average number of useful plants in 10×10 km grid cells was the highest of all bioregions, 67.3. In our study, the Amazon region had a relatively low proportion of surveyed and well surveyed cells, just over 34.5% and 23% in the case of all vascular plants and useful plants respectively. Survey coverage for useful plants of Amazonia was not adequate along both temperature and precipitation gradients. Our study suggests that only about 8% of the area has been transformed by human activities. This is one of the least accessible regions due to the reduced road network, which could explain poor sampling. Given a relatively low survey completeness and coverage in the Amazonian moist forest, species numbers in this bioregion are likely to have been underestimated by our study. This is reinforced by a relatively low number of species given the extent of the area sampled (S2.6 Fig in S2 File) and the high estimation of species richness for the whole Amazon [45]. Although the origin of the Amazonian flora was dated for many major clades back to the Late Cretaceous or Palaeocene, more recent landscape dynamics have led to high diversification rates in the last 5 million years, contributing to a high species diversity of the region [46].

**Caribbean region.** In our study, the region was subdivided into Caribbean páramo (the Santa Marta páramo), moist and dry forests, and Caribbean deserts and xeric shrublands.

Both moist and dry Caribbean forests had fewer vascular and useful plants than similar ecosystems in other regions, but they were also the smallest in size in the respective ecosystem categories. Mean richness of useful plants in the well surveyed area, however, was similar to that in moist and dry forests from other regions. In the forests of the Caribbean region, the percentage of well surveyed areas was lower than in other regions, 17.9–22.5.

The Santa Marta páramo is an ecoregion containing páramo vegetation above the treeline in the Sierra Nevada de Santa Marta mountain range on the Caribbean coast of Colombia. It is characterized by high diversity of habitats and high proportion of endemic species [47]. In our study, the Caribbean páramo had the smallest area of all bioregions (ca 1,700 km$^2$). The bioregion had the lowest number of vascular plants and useful plants. It was the only bioregion where 100% and 70.6% of all 10×10 km grid cells had observation records of all vascular and useful plants respectively. At the same time, the proportions of well surveyed cells were low, 23.5% and 8.3% for vascular and useful plants respectively. Even though we did not identify any areas transformed by human activities, the biodiversity of páramos is under threat from uncontrolled cattle and sheep grazing and wood extraction for fuel and building construction.

**Pacific region (Chocó).** In our project, the region was represented by the Pacific moist forest and mangroves. With 5,168 and 1,336 vascular plants and useful plants respectively, the Pacific moist forest was the third most species rich bioregion after the Andean and Pacific moist forests. The outstanding richness of the Pacific moist forests, which are part of the Chocó biodiversity hotspot, might have developed only in the last 5 million years once the landscape had changed into fully terrestrial [11]. The Chocó's flora has a great affinity with that of Central America. Given a recent development of the Chocó's landscape, it seems unlikely that the Andean uplift had played a major role in plant diversification [11]. In the well surveyed area, mean richness of vascular plants and useful plants was relatively high, 131.9 and 65.4; possibly a result of the Chocó being one of the wettest places on earth [48]. In the well surveyed area, survey completeness was just over 40% and 30% for vascular plants and useful plants, and survey coverage was particularly poor along the annual rainfall gradient.

## Potential uses of Colombia's vascular plants

In all bioregions, Medicines were the most common plant use, followed by Materials, Environmental uses, and Human food (Fig 3B). When the total numbers of species per use category were compared with the global World Checklist of Useful Plant Species [28], the proportions were similar. For example medicinal species, species associated with materials and environmentally useful species represented 76%, 38% and 32% respectively in Colombia and 66%, 34% and 22% globally.

There has been a strong recognition of the traditional knowledge and the importance of medicinal plants in Colombia that dates back to the 16th–18th centuries [49, 50]. However, the first scientific approach to the study of the plants and their uses in Colombia was probably applied by the Royal Botanical Expedition to New Granada (1783–1816) headed by José Celestino Mutis [51]; the expedition had preceded that of Humboldt [52]. In the last 30 years useful plants have been increasingly documented [47].

Several legislation and governmental strategies have been developed recently to improve medicinal plants knowledge, their protection and sustainable use [8]. Colombia's population not only commonly combine traditional and western medicine, but there is also a growing industry producing traditional medicines which has made the legislation necessary [8]. Globally, the demand for medicinal plants has increased in response to the outbreak of the SARS-CoV-2 virus thus creating new opportunities for the development of "green", integrative medicine [53]. The results of our study further confirm the potential of Colombia's vascular flora for the development of both traditional and western medicine.

Environmental uses were the third most common useful attribute across all bioregions. During the outbreak of the SARS-CoV-2 virus, the popularity of one particular type of environmental uses, ornamental, has considerably increased. Worldwide, cultivated plants in general and indoor vegetation in particular proved to be a source of mental support to those confined to their homes during the pandemic [54]. Thus, useful plants have a potential to help communities in challenging times.

After Fabaceaea (beans), Arecaceae (palms) is the second botanic family with most species used as materials in Colombia, and it has been suggested that this might be due to its well curated taxonomy [55]. Colombia not only has the highest number of useful palm species in South America but also the highest number of indigenous communities for which information has been collected [55]. This highlights the importance of combining reliable taxonomy with comprehensive ethnobotanical studies.

An improved knowledge of useful plants can directly contribute to achieving several UN Sustainable Development Goals (https://www.un.org/sustainabledevelopment/sustainable-development-goals/). For example, Goal 3 (Good health and wellbeing) can be supported by the increased use of medicinal plants. By developing sustainable management strategies for products derived from useful plants, both Goal 1 (No poverty) and Goal 12 (Responsible consumption and production) could also be assisted.

## Information gaps in Colombian species distribution record

Our results revealed several information gaps that limit our understanding of the distribution patterns of useful species and the role of the Colombian ecosystems in useful species supply. Those gaps were first of all revealed by our estimates of survey completeness and coverage. The proportion of well surveyed (i.e. with $\geq 25$ collection localities) grid cells was below 50% in all bioregions except for the Andean páramo (S2.10 Table in S2 File). The identified gap in collections could have a strong effect on the estimates of useful plant numbers. Collections should be improved particularly in the Savannas (Llanos), one of the bioregions with the

smallest surveyed area and with the lowest proportion of well surveyed cells. Likewise, the geographic coverage of collections across the Amazon and Llanos moist forests areas should be improved. Lastly, to improve survey completeness and coverage in the Caribbean páramos and the Caribbean deserts and xeric shrublands collection numbers should be increased.

The distribution of collection localities in several bioregions along temperature and precipitation gradients suggested inadequate survey coverage in environmental space, particularly in the drier and warmer areas (S2.3–S2.5 Figs in S2 File). The latter could undermine further efforts to predict distributions of useful species and their response to ongoing climate change [22]. When all Colombia's vascular plants were considered, the results were largely similar with the differences attributed to a larger size of all vascular plants dataset in comparison with the useful plants dataset.

While being a useful basis for comparison among bioregions, species to cell ratio (or cell based mean species richness) proved to be an unreliable estimate, when calculated for the whole study area. In our study, mean richness values across bioregions varied depending on the subset of cells in question, i.e. all cells in the study region, all surveyed or well surveyed grid cells (S2.5 and S2.6 Tables in S2 File). When using occurrence records in species distribution pattern assessments, we recommend reporting mean species richness for all categories of survey completeness.

Of the original list of 4200 useful species gathered for Colombia 3870 species had geographic records in GBIF and SIB, indicating that ca 8% of useful species do not have distribution data in globally available databases. While some additional records might be stored in local herbaria, and yet must be shared on online databases, the lack of records of some species could be due to their "non-native" nature. It has been reported for medicinal plants growing in Colombia, that only around 82% of them were native while the rest were introduced species [8]. The report of Bernal and colleagues was based on published studies where geographic coordinates have not always been recorded. In fact, introduced species are less recorded by biologists on average than native species [56]. It is therefore urgent to collect more records for the underrepresented useful native species but at the same time, to record the geographic locations of the introduced species. The latter is particularly relevant because the economic value of introduced species often helps them into a successful naturalisation [57], which could eventually lead to invasiveness, thus affecting local biodiversity and human wellbeing [58] (Pyšek et al., 2020).

## Better documented traditional knowledge and Colombian flora are key to improved knowledge about useful plants and their uses

High biocultural diversity [59] together with the astonishing numbers of plant species [60] makes Colombia a unique source of useful plants. The knowledge about useful plants can be improved by better documenting both traditional uses and floristic diversity.

Although the knowledge of Colombian indigenous communities has been reported in books and papers, there is still more work to do in this remit [7]. Traditional knowledge is held not only by indigenous communities, but also by Afro-Descendant Communities [61], people living in rural areas of Colombia such in Boyacá and Cundinamarca [13, 62] and people working in local markets in some cities in Colombia [63, 64]. Unfortunately, this knowledge has not been fully documented and the risk of losing it due to migration from rural to urban areas is high. Recent studies estimate that up to 90% of the population of Latin America and the Caribbean will live in cities by 2050 [65, 66]. Although Colombian people still grow useful species in their gardens, the proportion of native species decreases with residences with rural

origin to those with urban origin [67]. Therefore, there is an urgent need for better preservation of traditional knowledge.

In addition to traditional knowledge, better documentation and research of Colombia's flora is fundamental for the discovery of species that could be useful for people. Every year several new plant species are discovered in South America [2] and in Colombia, and some of those plants can have useful properties. In addition, some newly discovered plants can be resilient to the on-going climate change [68]. Therefore, unlocking plants' useful properties requires a concerted research effort from multiple disciplines. This concerns not only newly discovered species but also those at risk of extinction [69]. The latter group should be researched not only from ecological, but also from genetic or molecular perspectives in order to get a holistic view on speciation and extinction of useful plants [18]. Collecting high-quality geographic records is a first step in the effort to document flora useful for people.

## Conclusions

Our study confirmed the Andean region as the most plant diversity rich in Colombia, with the Andean moist forest being the most species rich of all bioregions described in our study. We also recorded high vascular plants and useful plant numbers in the Amazonian moist forest. While the Andean bioregions had a relatively high proportion of surveyed and well surveyed areas, the analysis of survey completeness and coverage suggests that sampling in Llanos and Amazon regions has been inadequate. These regions should become a priority in the future biodiversity surveys. On the other hand, the Andean páramos are the best sampled bioregion.

Consistently with global estimates, Medicines were the most common uses across all bioregions, followed by Materials, Environmental uses and Human food.

Our results indicate that sampling effort for useful plants of Colombia has been inadequate in the majority of the bioregions. This is supported by relatively low proportion of surveyed and well surveyed areas across bioregions, and inadequate survey coverage in environmental space.

## Supporting information

**S1 File. Colombia's flora: Data processing flow and results.**
(DOCX)

**S2 File. Additional tables and figures.**
(DOCX)

## Acknowledgments

We thank Carolina Castellanos, from the Instituto Alexander von Humboldt for providing data on the number of Colombian plant species and Robert Turner from Royal Botanic Gardens, Kew for his support with name matching.

This work is part of the Useful Plants and Fungi of Colombia (UPFC) project, delivered by the Royal Botanic Gardens, Kew (Kew), in partnership with the Alexander von Humboldt Biological Resources Research Institute (IAVH). The UPFC project aims to enhance nature's contribution to people in Colombia by 1) increasing and consolidating knowledge on Colombia's useful plants and fungi and making it accessible for the benefit of the society; 2) promoting a market for useful native species and their high-value natural products; and 3) encouraging the sustainable use of natural resources that protects the environment and enhances biodiversity.

## Author Contributions

**Conceptualization:** Carolina Tovar, Alexandre Monro, Justin Moat, Mauricio Diazgranados.

**Data curation:** Nadia Bystriakova, Carolina Tovar, Pablo Hendrigo, Julia Carretero, Germán Torres-Morales, Mauricio Diazgranados.

**Formal analysis:** Nadia Bystriakova, Carolina Tovar, Pablo Hendrigo.

**Funding acquisition:** Mauricio Diazgranados.

**Investigation:** Nadia Bystriakova.

**Methodology:** Nadia Bystriakova, Alexandre Monro.

**Project administration:** Mauricio Diazgranados.

**Resources:** Justin Moat.

**Supervision:** Carolina Tovar, Justin Moat.

**Visualization:** Nadia Bystriakova, Carolina Tovar.

**Writing – original draft:** Nadia Bystriakova, Carolina Tovar, Alexandre Monro, Pablo Hendrigo.

**Writing – review & editing:** Nadia Bystriakova, Carolina Tovar, Alexandre Monro, Justin Moat, Julia Carretero, Germán Torres-Morales, Mauricio Diazgranados.

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
