## [Decision Letter · Decision Letter 0]

14 Apr 2021

PONE-D-21-09140

Colombia's bioregions as a source of useful plants

PLOS ONE

Dear Dr. Bystriakova,

Thank you for submitting your manuscript to PLOS ONE. After careful consideration, we feel that it has merit but does not fully meet PLOS ONE’s publication criteria as it currently stands. Therefore, we invite you to submit a revised version of the manuscript that addresses the points raised during the review process.

We look forward to receiving your revised manuscript.

Kind regards,

Daniel de Paiva Silva, Ph.D.

Academic Editor

PLOS ONE

Journal Requirements:

3. We note that Figures 1 and 4 and Supporting Information Figures S2.1 and S2.2 in your submission contain map/satellite images which may be copyrighted.

a. You may seek permission from the original copyright holder of Figures 1 and 4 and Supporting Information Figures S2.1 and S2.2, to publish the content specifically under the CC BY 4.0 license. 

Reviewers' comments:

Reviewer's Responses to Questions

**Comments to the Author**

1. Is the manuscript technically sound, and do the data support the conclusions?

Reviewer #1: Yes

Reviewer #2: Yes

2. Has the statistical analysis been performed appropriately and rigorously? 

Reviewer #1: Yes

Reviewer #2: N/A

3. Have the authors made all data underlying the findings in their manuscript fully available?

Reviewer #1: Yes

Reviewer #2: Yes

4. Is the manuscript presented in an intelligible fashion and written in standard English?

Reviewer #1: Yes

Reviewer #2: Yes

5. Review Comments to the Author

Reviewer #1: Comments:

General comments:

In the work “Colombia's bioregions as a source of useful plants”, the authors assembled a dataset of georeferenced collection localities of all vascular plants and useful plants of Colombia, and put all point locality information in a map of Colombia’s bioregions, and they found Andean, Amazon, Pacific, Llanos and Caribbean moist forests have the highest numbers of all vascular plant species and useful species descendingly. The manuscript is well written and data analysis is supportive of their conclusions. I think, with more work like this, may be conducted in other areas in the world, we will have a more specific layout of the useful plants and detailed conservation plans of the plant species in the Anthropocene during which nearly all the plant species face a high extinction risk.

I am very happy to see such a work be published if they could address my following concerns:

1) It’s important to stress the importance of this or this kind of work. Collecting the information of useful plants in a specific area and showing the results publicly, I think, one of (those) aims to protect or conserve these valuable plant species, which should be addressed in the Introduction and Discussion. The authors may find it useful as a reference (Gao et al, Plant extinction excels plant speciation in the Anthropocene. BMC Plant Biology 2020, 20: 430. https://doi.org/10.1186/s12870-020-02646-3).

2) The authors said “traditional knowledge and the importance of medicinal plants in Colombia that dates back to the last 30 years” and “Colombia ethnobotanical studies on indigenous populations began in 1860 with the work of Florentino Vezga”, which probably not the truth. As I knew, Alexander von Humboldt has traveled to South America (or Colombia) and collected many useful plants and carried them to Europe which was definitely before 1860 (which highlighted in their Acknowledgements). The authors may find it useful to read Wulf’s book (The invention of nature: The adventures of Alexander von Humboldt, the lost hero of science (John Murray, London, UK, 2015)).

3) The authors explained the biodiversity of Pacific moist forests that “might have developed only in the last 5 million years once the landscape had changed into fully terrestrial”, which is very interesting. Can you expand it? (Including other bioregions of their manuscript) Linking plant evolutionary history and plant use is very crucial for our understanding of the current use scenarios of many plants. The authors may find it useful to refer Molina-Venegas’s recent work (Maximum levels of global phylogenetic diversity efficiently capture plant services for humankind. https://doi.org/10.1038/s41559-021-01414-2).

4) The authors indeed discussed their survey completeness and survey coverage of the useful plants, but not highlighted in the Discussion fully. The authors may find it useful to refer to a similar work “Ethnobotanical Knowledge Is Vastly Under-Documented in Northwestern South America” by Camara-Leret et al. (doi: 10.1371/journal.pone.0085794).

5) I strongly suggested moving Table S2.2 (Categories of plant uses following Diazgranados et al. (2020)) and Table S2.3 (Useful plants and their uses) to the main text. Because they are very crucial to understand some definitions used in this study. And one thing you should keep in mind, some plants are used both as Medicines and Human Food, or tri-usefulness (three or even more uses), which may be explained in the table caption.

6) In the discussion, they highlighted the importance of useful plants in promoting a bio-economy (reference 36 they cited), but the readers may be interested in the applications of these useful plants in achieving UN Sustainable Development Goals, especially in the face of COVID-19. For me, the useful plants in Colombia are key, at least, to some extent, in achieving Sustainable Development Goals 1 (No Poverty), 12 (Responsible Consumption and Production), 13 (Climate Action), and 15 (Life on Land). Can you expand it?

Minor comments:

Line 29: Formatting issue: gaps-,.

Line 203: Use abbreviations because you have already defined them (Line 157: annual mean temperature (ANMT) and total annual precipitation (ANP)).

Line 320: According to what? I believe you gave such an exact number (70%) must have based on some empirical study.

Line 436: 3870/4200 is a high proportion, not “only”.

Reviewer #2: There are some well-known problems with the used data, e.g. with GBIF, but the analyses seem to be sound and these data problems should not affect the general results of the study. So, I don't see any major technical issues.

6. PLOS authors have the option to publish the peer review history of their article (what does this mean?). If published, this will include your full peer review and any attached files.

Reviewer #1: **Yes: **Jianguo Gao

Reviewer #2: No

---

## [Author Response · Author response to Decision Letter 0]

27 May 2021

Response to specific editor comments.

 R: We have checked the style requirements as recommended.

R: We confirm that we will provide repository information for our data at acceptance.

3. We note that Figures 1 and 4 and Supporting Information Figures S2.1 and S2.2 in your submission contain map/satellite images which may be copyrighted.

R: We hereby confirm that the above mentioned figures are the authors' maps/figures and there is no copyrightable material.

R: We made the required corrections. 

Response to reviewer comments. 

We thank Dr Gao and one anonymous reviewer for their encouraging comments on our manuscript. Please, see our detailed response below. 

Reviewer 1

General comments:

In the work “Colombia's bioregions as a source of useful plants”, the authors assembled a dataset of georeferenced collection localities of all vascular plants and useful plants of Colombia, and put all point locality information in a map of Colombia’s bioregions, and they found Andean, Amazon, Pacific, Llanos and Caribbean moist forests have the highest numbers of all vascular plant species and useful species descendingly. The manuscript is well written and data analysis is supportive of their conclusions. I think, with more work like this, may be conducted in other areas in the world, we will have a more specific layout of the useful plants and detailed conservation plans of the plant species in the Anthropocene during which nearly all the plant species face a high extinction risk. I am very happy to see such a work be published if they could address my following concerns:

1) It’s important to stress the importance of this or this kind of work. Collecting the information of useful plants in a specific area and showing the results publicly, I think, one of (those) aims to protect or conserve these valuable plant species, which should be addressed in the Introduction and Discussion. The authors may find it useful as a reference (Gao et al, Plant extinction excels plant speciation in the Anthropocene. BMC Plant Biology 2020, 20: 430. https://doi.org/10.1186/s12870-020-02646-3).

R: As suggested, we made some amendments to Introduction and Discussion, and we also added the following citation to the list of references:

Gao J-G, Liu H, Wang N, Yang J, Zhang X-L. Plant extinction excels plant speciation in the Anthropocene. BMC Plant Biol. 2020;20: 430. doi:10.1186/s12870-020-02646-3

2) The authors said “traditional knowledge and the importance of medicinal plants in Colombia that dates back to the last 30 years” and “Colombia ethnobotanical studies on indigenous populations began in 1860 with the work of Florentino Vezga”, which probably not the truth. As I knew, Alexander von Humboldt has traveled to South America (or Colombia) and collected many useful plants and carried them to Europe which was definitely before 1860 (which highlighted in their Acknowledgements). The authors may find it useful to read Wulf’s book (The invention of nature: The adventures of Alexander von Humboldt, the lost hero of science (John Murray, London, UK, 2015)).

R: Thank you for the suggestion, we now have updated these sentences with Humboldt but also with other references from the sixteen and eighteen centuries.

3) The authors explained the biodiversity of Pacific moist forests that “might have developed only in the last 5 million years once the landscape had changed into fully terrestrial”, which is very interesting. Can you expand it? (Including other bioregions of their manuscript) Linking plant evolutionary history and plant use is very crucial for our understanding of the current use scenarios of many plants. The authors may find it useful to refer Molina-Venegas’s recent work (Maximum levels of global phylogenetic diversity efficiently capture plant services for humankind. https://doi.org/10.1038/s41559-021-01414-2).

R: We have now added information about the origin of the flora for all bioregions except for the Caribbean one for which this information was more difficult to obtain. However, we did not include a specific link with useful plants because we don’t have enough evidence for doing this in the region and we feel this is out of the scope of the paper.

4) The authors indeed discussed their survey completeness and survey coverage of the useful plants, but not highlighted in the Discussion fully. The authors may find it useful to refer to a similar work “Ethnobotanical Knowledge Is Vastly Under-Documented in Northwestern South America” by Camara-Leret et al. (doi: 10.1371/journal.pone.0085794).

R: The above article was cited in the earlier draft of the manuscript (number 7 on the list of references). We agree that our study has some similarities with the work of Camara-Laret et al., however, the methods and materials were very different, which makes a comparison of the studies rather difficult. In particular, our study used the records of botanical collections, which were not necessarily assembled with plant uses in mind. In fact, some botanists who gathered data on distribution of Colombia’s useful plants might not even be aware of their useful attributes. We therefore limit our discussion of survey completeness and coverage by the reference to the results based on the analysis of species distribution datasets (such as GBIF), without making direct comparisons with the results of ethnobotanical studies, which we feel are out of the scope of the present work. 

5) I strongly suggested moving Table S2.2 (Categories of plant uses following Diazgranados et al. (2020)) and Table S2.3 (Useful plants and their uses) to the main text. Because they are very crucial to understand some definitions used in this study. And one thing you should keep in mind, some plants are used both as Medicines and Human Food, or tri-usefulness (three or even more uses), which may be explained in the table caption.

R: As suggested, we moved Tables S2.2. and S2.3. to the main document (now Tables 1 and 2). We also edited the Table 2 caption as suggested.

6) In the discussion, they highlighted the importance of useful plants in promoting a bio-economy (reference 36 they cited), but the readers may be interested in the applications of these useful plants in achieving UN Sustainable Development Goals, especially in the face of COVID-19. For me, the useful plants in Colombia are key, at least, to some extent, in achieving Sustainable Development Goals 1 (No Poverty), 12 (Responsible Consumption and Production), 13 (Climate Action), and 15 (Life on Land). Can you expand it?

R: We already mentioned how medicinal plants have been highlighted under the current circumstances of the pandemic (l. 406-411). As suggested, we have added a small paragraph about the UN Sustainable Development goals (l.419-424).

Minor comments:

Line 29: Formatting issue: gaps-,.

R: Corrected.

Line 203: Use abbreviations because you have already defined them (Line 157: annual mean temperature (ANMT) and total annual precipitation (ANP)).

R: Corrected.

Line 320: According to what? I believe you gave such an exact number (70%) must have based on some empirical study.

R: A reference to table S2.3 added.

Line 436: 3870/4200 is a high proportion, not “only”.

R: We agree with the comment; “only” has been deleted. 

Reviewer 2

There are some well-known problems with the used data, e.g. with GBIF, but the analyses seem to be sound and these data problems should not affect the general results of the study. So, I don't see any major technical issues.

---

## [Decision Letter · Decision Letter 1]

14 Jun 2021

PONE-D-21-09140R1

Colombia's bioregions as a source of useful plants

PLOS ONE

Dear Dr. Bystriakova,

Thank you for submitting your manuscript to PLOS ONE. After careful consideration, we feel that it has merit but does not fully meet PLOS ONE’s publication criteria as it currently stands. Therefore, we invite you to submit a revised version of the manuscript that addresses the points raised during the review process.

We look forward to receiving your revised manuscript.

Kind regards,

Daniel de Paiva Silva, Ph.D.

Academic Editor

PLOS ONE

Journal Requirements:

Additional Editor Comments (if provided):

Dear Bystriakova et al.,

Congratulations, your manuscript is almost there!

After this second round of reviews, one previous reviewer decided for the acceptance, whereas the second new one indicated minor improvements to be performed.

Please resubmit the final version of your MS in a one-month period (12th July, 2021) or at your earliest convenience. During the resubmission process, do not forget to prepare a rebuttal letter informing of the performed changes.

Sincerely,

Daniel Silva, PhD

Reviewers' comments:

Reviewer's Responses to Questions

**Comments to the Author**

1. If the authors have adequately addressed your comments raised in a previous round of review and you feel that this manuscript is now acceptable for publication, you may indicate that here to bypass the “Comments to the Author” section, enter your conflict of interest statement in the “Confidential to Editor” section, and submit your "Accept" recommendation.

Reviewer #1: All comments have been addressed

Reviewer #3: (No Response)

2. Is the manuscript technically sound, and do the data support the conclusions?

Reviewer #1: Yes

Reviewer #3: Yes

3. Has the statistical analysis been performed appropriately and rigorously? 

Reviewer #1: Yes

Reviewer #3: Yes

4. Have the authors made all data underlying the findings in their manuscript fully available?

Reviewer #1: Yes

Reviewer #3: Yes

5. Is the manuscript presented in an intelligible fashion and written in standard English?

Reviewer #1: Yes

Reviewer #3: Yes

6. Review Comments to the Author

Reviewer #1: (No Response)

Reviewer #3: I've been brought in to review this paper after one round of review has already been completed. I first read through the paper and then the response to reviewers letter, so that I could first make an opinion on the paper and then see if they have addressed previous criticism adequately.

Really I think this is an excellent paper, one which could provide a methodological template for future studies. The authors have been extremely thorough, not only in assembling and cleaning the data but also in analysing it. I only have minor suggestions. Had I been a first round reviewer, I might have made a couple of suggestions for alternative analyses that might provide some more perspectives on the sampling heterogeneity that affects the dataset. However the analyses shown in the paper are definitely sufficient, and I don't think it fair for a new reviewer to be brought in on the second round of reviews and start demanding new analyses; they would certainly not be essential to the paper which stands very well on its own merit. I will therefore only suggest that it might be worth showing, maybe as a supplementary figure, a scatterplot directly comparing within each bioregion, number of species to number of sampled/well sampled grid cells. Given the number of bioregions a correlation test would probably not be much use, but such a figure would provide quite a clear illustration a) of how strong the relationship is (and therefore potentially how much of the data signal is sampling), and which bioregions have substantially more/less data than would be expected given the extent of sampling. However I do recognise that the authors have already done substantial work on this paper and have definitely done enough to show their conclusions are valid, and so I won't insist on this and I think the paper is perfectly acceptable as is, barring a couple of wording and grammar quibbles:

Abstract line 29: “…contained the largest numbers of useful plants.” Phrasing is ambiguous; number of plants could be taken to mean absolute abundances. Maybe specify you mean largest number of useful plant *species*.

Discussion, line 290: “…with 22.7% of well surveyed cells savanna had the lowest survey completeness.” Need comma after ‘cells’

7. PLOS authors have the option to publish the peer review history of their article (what does this mean?). If published, this will include your full peer review and any attached files.

Reviewer #1: **Yes: **Jianguo Gao

Reviewer #3: No

---

## [Author Response · Author response to Decision Letter 1]

7 Jul 2021

Dear Editor,

Please, see our response to specific reviewer comments below. 

We thank an anonymous reviewer for their encouraging comments on our manuscript. Please, see our detailed response below. 

Reviewer #1: (No Response)

Reviewer #3: I've been brought in to review this paper after one round of review has already been completed. I first read through the paper and then the response to reviewers letter, so that I could first make an opinion on the paper and then see if they have addressed previous criticism adequately.

Really I think this is an excellent paper, one which could provide a methodological template for future studies. The authors have been extremely thorough, not only in assembling and cleaning the data but also in analysing it. I only have minor suggestions. Had I been a first round reviewer, I might have made a couple of suggestions for alternative analyses that might provide some more perspectives on the sampling heterogeneity that affects the dataset. However the analyses shown in the paper are definitely sufficient, and I don't think it fair for a new reviewer to be brought in on the second round of reviews and start demanding new analyses; they would certainly not be essential to the paper which stands very well on its own merit. I will therefore only suggest that it might be worth showing, maybe as a supplementary figure, a scatterplot directly comparing within each bioregion, number of species to number of sampled/well sampled grid cells. Given the number of bioregions a correlation test would probably not be much use, but such a figure would provide quite a clear illustration a) of how strong the relationship is (and therefore potentially how much of the data signal is sampling), and which bioregions have substantially more/less data than would be expected given the extent of sampling. However I do recognise that the authors have already done substantial work on this paper and have definitely done enough to show their conclusions are valid, and so I won't insist on this and I think the paper is perfectly acceptable as is, barring a couple of wording and grammar quibbles:

R: As suggested by Reviewer, we added a supplementary figure (Fig. S2.6) comparing all vascular plant and useful plant species numbers to numbers of sampled grid cells within each bioregion. The same test for well sampled grid cells would require a much more complicated and time consuming data analysis. As the latter was not planned at the initial stages of the manuscript preparation, an additional analysis would mean substantial changes to Materials&Methods and Results sections, while being not essential to the paper as a whole. When a new figure S2.6 is considered, Amazon moist forests and Llanos savannas appear to be particularly data deficient relative to other regions; thus the scatterplots did not reveal any strong outliers in addition to those already discussed in the text of the manuscript. Therefore we did not make any changes to Discussion apart from making a brief reference to the figure.

Abstract line 29: “…contained the largest numbers of useful plants.” Phrasing is ambiguous; number of plants could be taken to mean absolute abundances. Maybe specify you mean largest number of useful plant *species*.

R: Changed as suggested.

Discussion, line 290: “…with 22.7% of well surveyed cells savanna had the lowest survey completeness.” Need comma after ‘cells’

R: Corrected.

---

## [Decision Letter · Decision Letter 2]

9 Aug 2021

Colombia's bioregions as a source of useful plants

PONE-D-21-09140R2

Dear Dr. Bystriakova,

We’re pleased to inform you that your manuscript has been judged scientifically suitable for publication and will be formally accepted for publication once it meets all outstanding technical requirements.

Kind regards,

Daniel de Paiva Silva, Ph.D.

Academic Editor

PLOS ONE

Additional Editor Comments (optional):

Dear Bystriakova et al.,

After another review around, I am pleased to inform you that your study has been formally accepted for publication in PLoS One!

Congratulations,

Daniel Silva, Ph.D.

Reviewers' comments:

Reviewer's Responses to Questions

**Comments to the Author**

1. If the authors have adequately addressed your comments raised in a previous round of review and you feel that this manuscript is now acceptable for publication, you may indicate that here to bypass the “Comments to the Author” section, enter your conflict of interest statement in the “Confidential to Editor” section, and submit your "Accept" recommendation.

Reviewer #1: All comments have been addressed

Reviewer #3: All comments have been addressed

2. Is the manuscript technically sound, and do the data support the conclusions?

Reviewer #1: Yes

Reviewer #3: Yes

3. Has the statistical analysis been performed appropriately and rigorously? 

Reviewer #1: Yes

Reviewer #3: Yes

4. Have the authors made all data underlying the findings in their manuscript fully available?

Reviewer #1: Yes

Reviewer #3: Yes

5. Is the manuscript presented in an intelligible fashion and written in standard English?

Reviewer #1: Yes

Reviewer #3: Yes

6. Review Comments to the Author

Reviewer #1: (No Response)

Reviewer #3: I am extremely sorry for taking so long to review this paper again. I think the authors have done an excellent job in addressing reviewer comments and I think the paper is good to go. I noticed two things that should be addressed in the proofing stage:

On line 491, the numbered reference [58] is followed by a name-in-brackets citation (Pyšek et al., 2020).

On line 525: "Consistently with global estimates, Medicines were the most common..." Consistently should be Consistent

I congratulate the authors on an excellent paper and I look forward to seeing it out.

7. PLOS authors have the option to publish the peer review history of their article (what does this mean?). If published, this will include your full peer review and any attached files.

Reviewer #1: **Yes: **Jian-Guo Gao

Reviewer #3: **Yes: **Neil Brocklehurst

---

## [Editor Report · Acceptance letter]

20 Aug 2021

PONE-D-21-09140R2 

Colombia’s bioregions as a source of useful plants 

Dear Dr. Bystriakova:

I'm pleased to inform you that your manuscript has been deemed suitable for publication in PLOS ONE. Congratulations! Your manuscript is now with our production department. 

Kind regards, 

on behalf of

Dr. Daniel de Paiva Silva 

Academic Editor

PLOS ONE